# Impact of Industrial Practices on the Microbial and Quality Attributes of Fresh Vacuum-Packed Lamb Joints

**DOI:** 10.3390/foods11131850

**Published:** 2022-06-23

**Authors:** María de Alba, Catherine M. Burgess, Katie Pollard, Camila Perussello, Jesús M. Frías-Celayeta, Des Walsh, Joan Carroll, Emily Crofton, Carol Griffin, Cristina Botinestean, Geraldine Duffy

**Affiliations:** 1Food Safety Department, Teagasc Food Research Centre, Ashtown, D15 KN3K Dublin, Ireland; mariadealbaortega@hotmail.com (M.d.A.); kaye.burgess@teagasc.ie (C.M.B.); kpollard@live.ie (K.P.); des.walsh@teagasc.ie (D.W.); joan.carroll@teagasc.ie (J.C.); 2Environmental Sustainability and Health Institute, Technological University Dublin, City Campus, D07 H6K8 Dublin, Ireland; camila.perussello@tudublin.ie (C.P.); jesus.frias@tudublin.ie (J.M.F.-C.); 3Food Quality and Sensory Science Department, Teagasc Food Research Centre, Ashtown, D15 KN3K Dublin, Ireland; emily.crofton@teagasc.ie; 4Food Industry Development Department, Teagasc Food Research Centre, Ashtown, D15 KN3K Dublin, Ireland; carol.griffin@teagasc.ie (C.G.); cristina.botinestean@teagasc.ie (C.B.)

**Keywords:** fleece, lamb carcass, chilling length, microbial load, meat quality, shelf-life

## Abstract

The impact of different industrial practices at lamb export abattoirs in Ireland on the microbial and quality attributes of fresh vacuum-packed (VP) lamb leg joints, including Clean Livestock Policy (CLP), fleece clipping, carcass chilling times and vacuum pack storage, at typical chill and retail display temperatures was investigated. Five separate slaughter batches of lamb (ranging in size from 38 to 60 lambs) were followed at two lamb export plants over a two-year period, accounting for seasonal variation. In general, fleece clipping resulted in significantly lower microbial contamination on the fleece than the use of CLP alone. Lamb from carcasses chilled for 24 h had significantly lower psychrophilic total viable counts and *Brochothrix thermosphacta* and pseudomonad counts than carcasses chilled for 72 h. Following vacuum-packed (VP) storage of meat from these carcasses at 1.7 ± 1.6 °C for 23 days in the meat plant followed by retail display at 3.9 ± 1.7 °C (up to day 50), the dominant microorganisms were lactic acid bacteria, *Br. thermosphacta*, Enterobacteriaceae and pseudomonads, and all had reached maximum population density by storage day 34. Aligned with this, after day 34, the quality of the raw meat samples also continued to deteriorate, with off-odours and colour changes developing. While the mean values for cooked meat eating quality attributes did not change significantly over the VP storage period, high variability in many attributes, including off-flavours and off-odours, were noted for lamb meat from all storage times, highlighting inconsistences in lamb quality within and between slaughter batches.

## 1. Introduction

There are many attributes which govern the shelf-life and consumer acceptability of vacuum-packed (VP) fresh lamb joints, including total microbial load and composition, colour, odour, flavour and tenderness [1,2]. The industrial practices along the lamb processing chain will impact these traits and the shelf-life of the primal cut.

In particular, it is recognised that the microbial load and composition on the lamb at the time of vacuum packaging is a key factor in determining the shelf-life achieved [2] and that the microbial profile at this stage of the chain is reflective of the cross-contamination of the carcass and meat cuts, which has occurred during slaughter, carcass dressing, chilling and boning-out operations. A significant risk factor for carcass contamination is the microbial load on the fleece of the sheep presented for slaughter, and the fleece-removal operation poses a high risk for microbial transfer. Most fleece contamination is of faecal origin, with some originating from the farm environment [3]. The prevailing weather conditions and season may significantly impact visual cleanliness of pasture-based lamb, while additional cross-contamination between sheep may occur during transportation, especially if the density of animals in the trucks is high [4]. Many countries, including Ireland, have a Clean Livestock Policy (CLP) system to support the management of dirty animals and reduce the risk of microbial contamination into the slaughter plant and subsequent carcasses contamination [5].

Aerial contamination in the plant and surface contamination on equipment and surfaces can also impact the carcass microbial load and profile [6]. The carcass chilling regime, including the temperature, air speed, relative humidity and duration of chilling, will impact both the carcass microbial profile and the lamb meat sensory qualities [7,8,9].

Once the lamb is in the vacuum pack, and as chilled storage under VP progresses, the dominant microorganisms change from a mainly aerobic population on the carcass to an anaerobic profile, including psychrotrophic LAB [10], psychrotrophic Enterobacteriaceae, *Brochothrix thermosphacta*, *Shewanella putrefaciens* and psychrophilic *Clostridium* spp. [11,12]. The lag phase, growth rate and the spoilage potential of these microorganisms in the VP lamb are governed by the storage temperature, pH of the meat and the partial pressure of O_2_ and CO_2_ [13]. Meat with a higher pH, such as lamb (≥5.8), will support faster growth of spoilage-related bacteria.

Changes in VP lamb appearance, off-odour and off-flavours are also related to the storage temperature, oxygen levels and the microbial composition and load [1,14]. Fresh VP lamb is known for the development of off-odours. These may be confinement odours, which disappear rapidly when the lamb is removed from the packaging, or they can be more permanent off-odours, ranging from slightly sour to dairy-like, which are produced by different spoilage microorganisms [13]. There are, however, gaps in knowledge on how off-odours may be impacted by industrial practices and on their inter-relationship with the changing microbial load and profile during VP storage.

Colour is an attribute strongly associated with consumer acceptability and can be a key shelf-life limiting factor. In VP meat, if there is residual oxygen at the time of packing or oxygen transmission into the pack during storage [15,16], the meat pigment myoglobin can be oxidised to metmyoglobin, giving a brown colour unacceptable to consumers. Colour changes are known to be highly related to the VP storage conditions, including temperature and oxygen levels. However, much of the data on meat quality attributes are from laboratory or pilot plant models, and there is limited information on how they are impacted by variations under industrial operating conditions. Additionally, the microbial data used to predict spoilage are often limited to total viable count (TVC) data rather than specific meat-spoilage-related microorganisms.

As industrial practices impact heavily on shelf-life outcome, the objective of this study was to track lamb at two Irish export abattoirs to study the effects of different industrial practices in Ireland, including the CLP, fleece clipping, carcass chilling times (24 h versus 72 h) and vacuum-packed storage at typical chill and retail temperatures, on the microbial and quality attributes of derived lamb carcasses and fresh vacuum-packed (VP) lamb leg joints. The aim was to build up a baseline of data on how different industrial practices in export lamb plants impact and interact on the attributes of fresh VP lamb shelf-life.

## 2. Materials and Methods

### 2.1. Lamb Sampling

Lamb batches, designated as CLP Category A or B [5], were selected at two commercial lamb export abattoirs. The CLP scoring is carried out in the lairage area, and is a three-category visual system, which requires food business operators at slaughtering establishments to categorise sheep as A, Satisfactory; B, Acceptable or C, Unacceptable. During the trial period, lambs scoring as CLP Category C fleece were considered to be of unacceptable cleanliness and could not proceed from lairage to be slaughtered, and they were thus not included in the present study.

Lambs at both plants were slaughtered at similar line speeds of 300 to 310 animals per hour. Between September 2017 and October 2019, sampling trips (*n* = 5 slaughter batches) were made to lamb abattoir 1, batch 1 (49 lamb, CLP B), batch 2 (56 lamb, CLP A) and batch 3 (60 lamb, CLP B), and at lamb abattoir 2, batch 4 (34 lamb, CLP A) and batch 5 (38 lamb, CLP B). The ear tag and carcass number from each animal were recorded for the collection of metadata (weight, herd, fat class) of each animal.

### 2.2. Fleece Swabbing

At abattoir 1, the fleece was clipped along the animal’s midline region, immediately after killing and before fleece removal. Following clipping, a sterile sampling sponge (TS/15-YCS, Technical Service Consultants Ltd., Heywood, Lancashire, UK), 10 cm × 10 cm and 10 mm thick; made of yellow polyurethane foam; pre-moistened with Maximum Recovery Diluent 0.1% peptone and 0.85% NaCl, MRD (Oxoid, Basingstoke, Hampshire, UK), was used to swab an area of 200 cm^2^ (2 × 100 cm^2^) along the mid-line region of the animal. After swabbing, the loaded sponge was withdrawn into the reverted bag [17]. At abattoir 2, the fleece was not clipped (as per normal procedure at the plant) and was swabbed as described above. The swabs were microbiologically analysed as described in Section 2.6. In both abattoirs, the fleece was pulled from the shoulders manually, and its removal was completed with a fleece puller.

### 2.3. Carcass Chilling

At both abattoirs, after dressing and evisceration and before chilling, the left side of all carcasses from the same batch was swabbed at four different sites (flank, lateral thorax, breast and brisket; 100 cm^2^ each; EC Decision 2001/471/EC), covering a total area of 400 cm^2^, and pooled. Carcass swabbing was carried out with a carcass sampling sponge as described above. All pooled swabs were returned to the laboratory under chilled conditions for microbiological analysis, as described in Section 2.6. After swabbing, all carcasses from the same batch were moved into the chill room, and air temperature was continuously monitored using data loggers (Model Easylog, Lascar Electronics, Whiteparish, Wiltshire, UK). Carcass surface temperature was measured on day 2 and day 5 using an Infra-Red thermometer (Model 810; Testo, Alton, Hampshire, UK). Carcass core temperature was measured in the inner leg on day 2 or on day 5 in the chilling room using a thermometer (Model HI 98509; Hanna Instruments, Woonsocket, RI, USA).

On day 1 (D1), 24 h post-slaughter, designated as T1, half of the carcasses from the same batch in the chill were selected at random, and the pH was measured in the leg muscle using a pH knife probe (Model Eutech pH 150; Thermo Scientific, Waltham, MA, USA). The previously un-swabbed right side of those carcasses was swabbed as above.

On day 3 (D3) (72 h post-slaughter, designated as T2), the pH of remaining carcasses from the same batch in chill was measured, and the un-swabbed right side of those carcasses was swabbed as above. All pooled swabs were returned to the lab under chilled conditions for microbiological analysis.

### 2.4. Aerial Sampling

Air sampling was carried out on D0 (killing day) in the fleece pulling area, evisceration area and chill room and on D1 and D3 post-slaughter, in the chill room and boning hall before and during carcass boning-out using a plate sedimentation method with placement of duplicate agar plates (no lids) exposed for one hour, after which the lids were replaced and the plates brought to the lab for microbiological analysis. The plates included Plate Count Agar (PCA, Oxoid), Violet Red Bile Glucose agar (VRBGA) plates (Oxoid) and Pseudomonas Agar Base (Oxoid) supplemented with Cetrimide Fucidin Cephalosporin (CFC) selective agar supplement (Oxoid); Streptomycin-thallous acetate-actidione agar base (STAA; Oxoid) containing STAA selective supplement (Oxoid) and de Man Rogosa Sharpe (MRS, Oxoid). Incubation conditions were as described in Section 2.6.

### 2.5. Boning-Out Lamb Leg Joints

On D1, the T1 half carcasses from each batch were moved into the boning hall, and the lamb hind legs were removed and cut into two joints (four joints obtained per carcass), with the bone left in the joints (bone-in). Each joint was placed in a vacuum pouch bag of 90 µm thickness and Oxygen transmission rate (OTR) of 11.02 cm^3^m^−2^day^−1^ (Sealed Air CRYOVAC^®^ Food Packaging Systems, Charlotte, NC, USA), and shank caps were used where appropriate to avoid leakers, and the samples were vacuum packed (VP) as joints. At both abattoirs, a CRYOVAC^®^VS95TS automatic belt vacuum packaging machine (Sealed Air CRYOVAC^®^ Food Packaging Systems, Charlotte, NC, USA) was used to vacuum pack the joints. All lamb joints were placed inside trays with data loggers to monitor the temperature, and transferred to the chill room for storage at the meat plant. This was repeated on D3 for the remaining carcasses from the same batch (T2).

Before and during carcass boning-out on D1 or D3, surface swabs, pre-moistened with a neutralising buffer (Envirostik kit, Technical Service Consultants Ltd., UK) (*n* = 10), were used to swab surfaces (100 cm^2^) of the cutting boards and conveyor belts. Swabs were returned to the lab under chilled conditions for microbiological analysis.

The VP lamb was stored at the abattoirs until D 23. Lamb leg joints (*n* = 3) from T1 and T2 were collected at random on days 1, 3, 6, 7, 8, 13, 15, 20, 21 and 22 and transported to Teagasc Food Research Centre, Ashtown (Dublin, Ireland), under refrigeration conditions. On D23, all lamb leg joints which remained at the abattoirs were collected, transported to Teagasc under refrigeration conditions, placed into retail display cabinets (Capital Galaxy G14 S/S Multideck with 2 shelves, Cross Refrigeration, Dublin, Ireland) and selected at random (*n* = 3) from T1 and T2 to be tested. Data loggers (Model Easylog, Lascar Electronics, UK) were used to monitor the temperature during transport and through retail display until D50. Samples were tested for their microbiological profile, visual colour and odour, thiobarbituric acid reactive substances (TBARS) and eating quality, described below.

### 2.6. Microbiological Analyses

In the laboratory, 40 mL of MRD was added to each fleece swab, each pair of pooled carcass swabs and to each environmental swab. The samples were homogenized in a stomacher (IUL, Barcelona, Spain), and serial dilutions were prepared in MRD.

Mesophilic and psychrophilic aerobic colony counts were determined on duplicate PCA plates (3M™ Petrifilm™ Aerobic, TVC), incubated at 30 °C for 48 h and 6.5 °C for 10 days, respectively. Total Enterobacteriaceae counts (TEC) were determined on duplicate VRBGA plates (3M™ Petrifilm™ Enterobacteriaceae) incubated at 37 °C for 24 h. *Pseudomonas* spp. was determined on duplicate plates of *Pseudomonas* Agar Base (Oxoid) supplemented with CFC selective agar supplement (Oxoid) incubated at 30 °C for 48 h. *Br. thermosphacta* was determined on duplicate plates of STAA (Oxoid) containing STAA selective supplement (Oxoid), incubated at 25 °C for 48 h. LAB was determined on duplicate plates of MRS (Oxoid) incubated at 30 °C for 72 h.

### 2.7. Microbiological Analyses of Meat Samples

The whole lamb leg joint surface areas were calculated in order to allow for determination of the cfu per unit area of meat. Joints were transferred to sterile bags and were shaken in 500 mL of MRD for approximately 90 s. Serial dilutions were prepared in MRD, and microbial determinations were carried out as described in Section 2.6.

### 2.8. Visual Colour and Odour Assessment of Raw Samples

To allow direct comparison of quality attributes on the same lamb joint that was microbiologically analysed, the raw meat colour was also measured through the package by a Hunter Lab colorimeter (Ultra Scan XE) for redness (CIE a* values), yellowness (CIE b* values) and lightness (CIE L* values). The raw meat samples were also assessed for visual colour and odour in the laboratory by four trained panellists, staff of Teagasc Food Research Centre, Ashtown (Dublin, Ireland). This supported the eating quality analysis of cooked samples from the same batch by an additional full-sized trained panel of 8 (Section 2.10). The colour of the meat was assessed through the packaging using a 5-point scale, where 1 = bright purple-red and 5 = brown [18]. Any score ≥ 3 was considered unacceptable. Off-odour was assessed by the panellists immediately after opening the packaging and re-sniffed after 30 min. Odour was scored on an 8-point scale, where 1 = none and 8 = very strong. Any score ≥ 4 was considered unacceptable. Finally, an overall quality score was assigned to the lamb using a 10-point scale, where 1 = reject and 10 = match. Any score ≤ 5 was considered unacceptable [19].

### 2.9. Lipid Oxidation

Secondary lipid oxidation products were estimated as thiobarbituric acid reactive substances (TBARS) according to Moran and colleagues [20]. Calibration was performed using a standard curve of 1,1,3,3,-tetraethoxypropane (TEP) solution, thus yielding malondialdehyde (MDA). Absorbance was measured at 532 nm (Shimadzu UV-1700, Columbia, MD, USA). Results were expressed as mg MDA kg^−1^ meat.

### 2.10. Eating Quality

Joints from T1 and T2 sampled at days 27, 28, 31, 34, 36, 38, 42, 44, 48 and 50 were frozen and stored at −20 °C until sensory analysis. An eight member trained external sensory panel [21], with a minimum of 2 years meat profiling experience, evaluated lamb joints at the Sensory Science Suite at Teagasc Food Research Centre, Ashtown (Dublin, Ireland). At the beginning of each session, panellists received a warm-up sample to reduce first order bias and to enhance calibration across panellists before sensory assessment. Lamb joints were cooked at 180 °C in a Rational Combi-Oven until an internal temperature of 71 °C was reached. Cooked lamb joints were cut into portions of 1.27 cm × 1.27 cm × 2.54 cm thickness, ensuring no gristle was present. Panellists assessed four portions of meat from each lamb joint: 2 × cubes of the crispy outer skin/rind and 2 × cubes of the internal meat without the outer skin/rind. The crispy outer skin/rind and the inner meat of the lamb joint were assessed separately. Data were captured using Compusense© Cloud software (Guelph, ON, Canada). Samples were assessed for off-odour, tenderness, juiciness, off-flavour and lamb flavour intensity. Off-odour and off-flavour were scored on a 5-point scale, where 1 = no off-odour/off-flavour and 5 = recognizable off-odour/off-flavour. Off-odour and off-flavour scores ≥ 4 were deemed to be unacceptable. Tenderness, juiciness and lamb flavour intensity were scored on a 10-point scale, where 1 = not tender, not juicy and no lamb flavour and 10 = very tender, very juicy and very strong lamb flavour. Samples were presented to panellists in a monadic sequential order under blind conditions, following an experimental design balanced for order and carry-over effects. Sensory evaluation took place in individual testing booths under red filtered light. Unsalted crackers and filtered tap water were used as palate cleansers during tastings.

### 2.11. Statistical Analysis

In order to study the effect of typical industrial practices on lamb microbiology and quality attributes and to consider the influence of variability within and between batches, linear mixed effect models with random intercepts were built, followed by post-hoc comparisons. The models used in each of the results are outlined below using Wilkinson and Rogers [22] notation:

Fleece microbial counts
(1)$$\log_{10}\left(fleece\right)\sim microorganism:CLP:clipping+\left(1|trial/carcass\right)$$

Aerial microbial counts in the fleece pull area.
(2)$$\;\log_{10}\left(aerial\right)\sim microorganism:CLP+(1|trial)$$

Microbial counts on derived pre-chill and post-chill carcasses.
(3)$$\;\log_{10}\left(carcass\right)\sim microorganism:CLP:stage+\left(1|trial/carcass\right)$$

Aerial microbial counts in the evisceration area, chill room, and boning hall.
(4)$$\;\log_{10}\left(aerial\right)\sim microorganism:CLP:post.slaughter.day+(1|trial)$$

Microbial counts on the boning hall surfaces.
(5)$$\;\log_{10}\left(surface\right)\sim microorganism:CLP:boning.out.day+(1|trial)\;$$
where the dependent variables on the left-hand side of the equations were expressed as cfu cm^−2^ counts on the fleece and carcass swabs, the aerial microbial counts in different processing areas were expressed as cfu m^−2^h^−1^ and the surface counts were expressed as cfu cm^−2^. All counts were modelled using a log_10_ transformation. The independent variables on the right hand side of the equations were all qualitative factors: microorganism indicated the bacterial groups tested (TVC-mesophilic and psychrophilic, Enterobacteriaceae, *Pseudomonas* spp., *Br. thermosphacta* and LAB); CLP indicated the CLP Categories A or B; clipping indicated the practice of clipping the fleece along the animal’s midline region (clipping or no-clipping); stage (pre-chill and post-chill) indicated the timing of the carcass swab; post.slaughter.day was a factor that indicated D0, D1 or D3 post-slaughter and boning.out.day indicated the day of boning-out (D1 or D3 post-slaughter). Finally, the random intercept effects trial (anonymised) and, where appropriate, the nested effect carcass (carcass number) within trial indicated the experimental batch conducted and the individual animal within that trial being evaluated.

Linear mixed effect models were fitted using the ‘lme4′ library [23] of the statistical software R [24]. The ‘lsmeans’ library [25] was used to conduct post hoc test via Satterthwaite’s degrees of freedom method [26] and with the Kenward–Roger degrees of freedom method on the mixed effect model marginal means and produced compact letter display tables for those tests.

Sensory scores of raw meat quality attributes of lamb joints were summarized by each CLP Category (A, B), treatment (T1, T2) and storage day, obtaining means and 0.05–0.95 quantiles for plotting. Sensory scores of eating quality attributes of lamb joints were summarized by crispy and non-crispy samples, treatment (T1, T2) and storage day, obtaining means and 0.05–0.95 quantiles for plotting. The Friedman’s test was performed on sensory data to determine the effect of treatment and storage day on the sensory quality of samples (Genstat version 18.1 for Windows, VSN International, Hemel Hempstead, UK). The data were averaged over sensory sessions and blocked by panellists to account for variation across panellists. Library ‘gtable’ was used to summarize statistical findings into tables. Libraries ‘ggplot2′ [27] and ‘dotwhisker’ were used to plot the results.

## 3. Results and Discussion

### 3.1. Fleece Management

Fleece management practices to reduce microbial contamination included a CLP scoring in both abattoir 1 and 2, while fleecing clipping was also carried out in abattoir 1. Slaughter batches scored as CLP A were more common in summer and CLP B and C in the autumn and winter months. However, during this trial period, lambs scored in the lairage as CLP C were not permitted to proceed to be slaughtered, and this will have mitigated the impact of the dirtiest lambs entering the plants, though the impact could not be directly measured in the study.

The results (Table 1) showed that clipping had a significant impact (*p* < 0.05), with significantly lower counts for TVC and spoilage microbial groups for all lambs scored as CLP B and on CLP A lamb for all microbial groups, except for LAB and Enterobacteriaceae, where no significant difference was noted.

In abattoir 2, when only the CLP was used, the scoring showed an impact for some microbial groups, with significantly lower counts for *Br. thermosphacta* and Enterobacteriaceae on CLP Category A fleece, compared to Category B (Table 1). As Enterobacteriaceae include many pathogenic bacteria, an impact of the CLP scoring on meat safety can be extrapolated. Byrne et al. [28] also reported a significant relationship between the visual cleanliness of fleeces and the total aerobic plate count, Enterobacteriaceae and coliforms counts on brisket, shoulder, flank and rump after pelt removal, irrespective of whether the fleece was wet or dry.

It is notable that the aerial microbial counts in the fleece pull area (Table 2) in abattoir 2, where no clipping was performed, were too numerous to count, while lower colony counts were obtained in abattoir 1, where fleece clipping was carried out. A CLP score of A or B had no significant impact on aerial counts. The TVC-mesophilic aerial counts detected in the fleece pull area were in line with those reported by [6] at the fleece pull stage at a commercial lamb plant. It is noted that the method of air sampling used was settle plates, as the use of an air sampler (impaction method) was not feasible in the industrial plants where the trials were performed. It could be expected that the microbial counts were underestimated by the approach used and biased towards heavier droplet deposition from the air. Okraszewska-Lasica et al. [6] reported a correlation of r^2^ = 0.77 between the settle plate counting approach used in this study and an air sampler (impaction method). Nonetheless, despite the limitation of the method, the results clearly demonstrate the potential for a carcass to be contaminated by the air in the plant and the impact of the fleece in introducing this contamination.

### 3.2. Carcass Chilling Time

In this study, both abattoir 1 and 2 carried out air chilling of the carcass for either 24 h (T1) or 72 h (T2), and the mean microbial data from both plants were used to look at the impact of this practice (Table 3). There was a very high level of variability in microbial counts noted both within and between slaughter batches, and at this stage, the CLP score (A or B) of the incoming lamb generally had no significant impact on microbial counts (log_10_cfu cm^−2^) on either pre-chill or post chill carcasses. It is noted that at the time of the study, only CLP A and CLP B scored lambs could proceed to slaughter, and if CLP C sheep had been included, the impact of this scheme would likely have been clearer.

The duration of carcass chilling (24 vs. 72 h) had a significant impact on the microbial spoilage groups monitored. Carcasses (from CLP B lambs) which had a longer chilling time (72 h) had significantly higher (*p* < 0.05) counts for TVC-psychrophilic, *Br. thermosphacta* and pseudomonads. This is probably reflective of the ability of psychrothrophic bacteria to recover and grow under carcass chilling conditions. The level of aerial microbial counts in the carcass chill room ranged from 2.18 to 3.69 log_10_ cfu m^−2^h^−1^ (Table 4) and such contamination levels could pose an ongoing risk for carcass contamination throughout the carcass chilling period. The study highlights that the length of the chilling time, prior to the bone-out of carcasses, is an important consideration when aiming to have lower microbial counts on meat at the time of bone out.

Lamb carcass’ ultimate pH values were very similar on D1 and on D3, with mean values of 5.93 ± 0.28 or 5.97 ± 0.30, respectively, which is in line with published pH data for lamb [29], which reported a variable pH range at 24 h of 5.31 to 5.87, with seasonal variations occurring. Lamb pH of ≥5.8, as noted in this study, would likely support faster growth of spoilage-causing bacteria during meat storage, and further research on managing the pH/temperature drop post-mortem during carcass chilling is warranted.

### 3.3. Boning-Out of Carcasses

The study investigated the levels of environmental microbial contamination in the boning hall, both on meat cutting surfaces and in the air. Mean microbial counts on boning hall surfaces at the time of carcass bone-out were as high as 2.5 log_10_ cfu cm^−2^, with some variability between the bone-out days (Figure 1). TVC-mesophilic counts were significantly higher (*p* < 0.05) on D3, while Enterobacteriaceae and *Brochothrix* were significantly higher (*p* < 0.05) on D1. Table 4 reviews the boning hall aerial mean microbial counts, which showed counts of up to 3.70 log_10_ cfu m^−2^h^−1^ for TVC mesophilic and 2.87 log_10_ cfu m^−2^h^−1^ for Enterobacteriaceae.

The aerial counts in the boning hall are reflective of the overall activity in that area [30], and many authors have reported that cutting boards are a major source for microbial contamination during boning, with higher Aerobic Plate Counts (APC) detected on lamb legs swabbed on the boning table after boning operations compared to the counts obtained on the chump area of the leg before entering the boning hall [31]. The data from this study clearly demonstrate that both the meat cutting surfaces and air are potential sources for cross contamination of the lamb meat being boned-out and that additional microbial control measures in the boning hall environment would be beneficial.

### 3.4. Vacuum-Packed Storage of Primals

In abattoir 1 and 2, the lamb joints were stored at in the meat plant at temperatures of 1.7 ± 1.6 °C until D23, followed by retail display storage at 4.6 ± 1.8 °C, up to D50, in order to reach the end of shelf life and assess when and what type of meat spoilage occurred.

Overall, there was high variability in the microbial counts (log_10_ cfu cm^−2^) on the lamb joints under these conditions, both within and between slaughter batches over the storage period (standard deviation (SD), 0.1 to 1.43 log_10_ cfu cm^−2^) (Figure 2), which likely masked the observation of inter-treatment effects from different carcass chilling times and CLP policies over the storage period. However, the data showed significantly lower (*p* < 0.05) counts for TVC Psychrophilic (2.00, 2.73 log_10_ cfu cm^−2^) on lamb derived from CLP B carcasses chilled for 24 h (T1) versus 72 h (T2), respectively. This followed the trend for the carcass microbial counts, showing the benefits of the shorter carcass chilling time.

Overall, at the time of packaging, the mean TVC-mesophilic counts ranged from 2.27 to 3.15 log_10_ cfu cm^−2^. These counts are similar to those previously reported [32] with TVC-mesophilic counts of 2.5–3.0 log_10_ cfu cm^−2^ noted on VP bone in hind-shank lamb at the time of packaging and counts ranging from 3.0 to 4.0 log_10_ cfu cm^−2^ on VP bone in lamb shoulders before packaging [33]. In this study, TVC-mesophilic counts had increased to 5.37–6.63 log_10_cfu cm^−2^ by D23 during storage in the meat plants and had reached final mean values of 7.17–7.6 log_10_ cfu cm^−2^ by D34. Gribble, Mills and Brightwell [34] reported similar increases in APC on VP fore-shank lamb samples, from 2.0–3.0 log_10_ cfu cm^−2^ prior to storage to 7.5–7.8 log_10_ cfu cm^−2^ after 9 weeks at 0 °C or 2 °C. A similar trend was noted for TVC-psychrophilic microbial counts (Figure 2). A lower VP storage temperature would be worthy of investigation by the lamb industry, with Kaur et al. [32] showing that the estimated growth rate of the TVC on VP bone in lamb hind-shanks stored at 8 °C (0.46 d^−1^) was approximately four-times faster than at −1.2 °C (0.12 d^−1^).

While there are no legislative microbial criteria for specific spoilage-related microorganisms on fresh meat in the EU, such criteria may likely be more useful in predicting shelf-life than the use of TVC data alone, and in this study, the growth of a number of meat spoilage-related microorganisms was investigated. At the time of packaging, LAB average counts ranged from 1.02 to 1.84 log_10_ cfu cm^−2^ and had increased to 4.24–6.15 log_10_ cfu cm^−2^ by D23 and to 6.30 to 6.78 log_10_ cfu cm^−2^ around D34. These levels are high and just below the levels (7 to 8 log_10_ cfu cm^−2^) which have reported to cause off-odour and off-flavours [35,36].

At the time, bone-out *Pseudomonas* counts ranged from 1.21 to 1.48 log_10_ cfu cm^−2^, and by D23, they had reached 3.20–4.66 log_10_ cfu cm^−2^ and 5.62–6.28 log_10_ cfu cm^−2^ by D34 in retail display storage. Although members of the *Pseudomonas* genus are commonly considered aerobic, they have previously been reported at levels of 3.4 to 5.2 log_10_ cfu g^−1^ on VP lamb primals stored at 0 °C or 5 °C, respectively, for 28 days [37]. As pseudomonads are generally considered to be aerobic, such results indicate the possible presence of some residual oxygen in the package or the transmission of oxygen into the packs during storage. It has been reported that pseudomonads are equipped with beneficial characteristics that allow them to withstand and adapt to harsh environmental conditions [38]. De Filippis and colleagues [39] demonstrated the presence of VP storage-adapted *P**. fragi* during meat spoilage. Further research using approaches such as metagenomic sequencing to characterise the pseudomonads present and the odours produced would be beneficial, as pseudomonads are known to produce volatile compounds on normal and high-pH meat, including a range of alkyl esters and a number of sulfur-containing compounds, including dimethyl sulfide [40], which is responsible for the “cabbage-like” odour of spoiled meat [41].

*Br. thermosphacta* counts ranged from 0.56 to 1.19 log_10_ cfu cm^−2^ on VP lamb joints at the time of packaging and increased to 4.26 or 5.56 log_10_ cfu cm^−2^ by D23 and to 4.78–5.24 log_10_ cfu cm^−2^ by D34 (retail display). Sheridan et al. [37] also reported *Br. thermosphacta* counts of 4.2 to 6.1 log_10_ cfu g^−1^ on VP lamb primals stored at 0 °C to 5 °C for 28 days. It has been noted that *Br. thermosphacta* may produce dairy odours when they reach levels of 10^6^ cfu g^−1^ [11,42].

On all joints, at the time of packaging, Enterobacteriacae counts ranged from 0.2 to 0.53 log_10_ cfu cm^−2^ and increased up to 2.69–4.11 log_10_ cfu cm^−2^ by D23 (in plant) and to 4.66–5.92 log_10_ cfu cm^−2^ by D34 in retail display. Some species of psychrotrophic Enterobacteriaceae are reported to produce sulphurous odours [35,43], and common thresholds for rejection by consumers are 3–4 log_10_ cfu cm^−2^ [34,44].

### 3.5. Raw Meat Quality Assessment

The study included a comparative analysis on the same lamb samples that were microbiologically analysed of the raw meat quality attributes, including colour, odour and overall quality (Figure 3), which were measured immediately on opening the pack and 30 min after by a trained panel. Off-odours on lamb joints from all treatments at the initial measurement and after 30 min scored between 1 (none) and 2 (trace, not sure) up to D34. From this storage day onwards, an increase in the off-odour was noted and was scored at an unacceptable level (≥4) by D48. The off-odours were still present after 30 min, indicating that true spoilage had occurred, rather than confinement odours, which usually dissipate within 30 min of opening of the vacuum pack [14]. By D34, all the spoilage-related microorganisms were at, or near their maximum population levels, and this may have contributed to the off-odours. As described above further research on the relationship between specific spoilage microorganisms and particular off-odours would be very useful if this issue is to be better managed by the lamb industry. The high variability in off-odours within and between slaughter batches was significant and also warrants further investigation.

Meat colour was measured by both the trained panel and a Hunter Lab colorimeter. Due to the high variability within and between batches, inter treatment differences from carcass chilling time and based on CLP policy were masked. Hunter data (a, b and L values) are not shown, as they showed the same trend as the visual scores.

For visual colour, initial ratings were variable, with score ratings generally between 2 (dull-purple red) and 3 (slightly brownish-red). Results after 30 min were more consistent with scores generally from 1 (bright-purple red) to 2 (dull-purple red) and, therefore, acceptable up to D50, albeit with some outlier joints reflecting the high variability. The change in colour that happened after being exposed to the air was due to the rebinding of atmospheric oxygen to heme iron of deoxymyoglobin (DMb) to form ferrous red oxymyoglobin (MbO_2_) [45]. From the study, it therefore appears that variability in colour within and between slaughter batches is a bigger issue than any particular industrial treatment effect.

The overall quality of lamb joints at the initial test and after 30 min was scored as acceptable (score > 5) up to D34. From D34 to D48, the lamb samples scored 4 (unacceptable) on the off-odour/off-flavour scale and 2 (reject) on the overall quality scale at the end of the storage period, with no treatment effect noted. This is a strong indication of a relationship with levels of specific spoilage microorganisms, most of which were at their maximum population at this time. There are opportunities for the lamb industry to have a more consistent shelf life by modifying the storage temperature and packaging environment.

Up to D34, the TBARS values on lamb samples, regardless of treatment, were lower than 1 mg MDA kg^−1^, which is considered the limiting threshold for acceptability in lamb [46]. However, in some lamb samples, TBARS levels of 1.23 mg MDA kg^−1^ were recorded at D38 (data not shown). This indicates that lipid oxidation products were being generated during prolonged storage and that oxygen was entering the vacuum packs, which had a recorded OTR of 11.02 cm^3^m^−2^day^−1^. This also correlates with the rancid off-odours noted on raw lamb samples from D34 onwards (Figure 3).

### 3.6. Eating Quality

Eating quality assessment of the cooked lamb joints (off-odour, tenderness, juiciness, off-flavour and lamb flavour) over 50 days storage from T1 (D1) or T2 (D3) are shown in Figure 4. The scores for off-odour and off-flavours on cooked lamb joints had an acceptable level (<4) until D50 (regardless of T1, T2). Mean tenderness scored between 7 and 8 from D27 until D50. The mean juiciness scores varied between 6 and 7, and mean lamb flavour scored between 6 and 8 during the 50 days. In general, all mean attribute scores from crispy and non-crispy samples followed a very similar pattern, with no significant differences between crispy and non-crispy samples. The eating quality of the crispy part (outer part) of the samples was assessed as the external surface, which has flavouring components and specific textural characteristics that attract consumers by giving an appealing appearance [47]. The overall trend showed no significant differences between T1 and T2 during storage.

There are limited data in the literature on eating quality assessment by a trained sensory panel of VP lamb joints stored at retail display conditions for 27 days within a study which has mimicked the industrial cold chain. However, the eating quality of lamb packed under vacuum skin packaging (VSP) and stored under retail display at 2 °C for up to 8 days has been studied [48], with tenderness, juiciness, liking of flavour and overall liking assessed by untrained consumers. The authors reported that VSP samples which received the highest scores were not affected by increasing display time.

## 4. Conclusions

This study on shelf-life of the VP bone in lamb leg joints followed under typical industrial conditions showed that the raw meat quality attributes, in particular, odour and, to lesser extent, colour, started to deteriorate at D34, and the raw meat was deemed unacceptable based on all these attributes at around D44. Interestingly, the cooked meat eating quality of the lamb joints did not change significantly over the storage period. The cause of the deterioration in the raw meat quality attributes from D34 onwards was most likely related to the levels of spoilage-related microorganisms, all of which were at or near maximum population at this time, including Enterobacteriaceae (4.66 to 5.92 log_10_ cfu cm^−2^), LAB (6.30 to 6.78 log_10_ cfu cm^−2^), *Br. thermosphacta* (4.78 to 5.24 log_10_ cfu cm^−2^) and pseudomonads (5.62 to 6.28 log_10_ cfu cm^−2^). The storage temperature (from 1.7 ± 1.6 °C up to D2 and from 4.6 ± 1.8 °C up to D50, respectively), with likely some oxygen transmission into the packs during storage, will have supported their proliferation to these threshold spoilage levels. Microbiological counts at the time of packing could be reduced by sourcing Cleaner CLP A scored lambs and by clipping of the fleece, combined with reducing the length of carcass chilling to 24 h and measures to reduce cross-contamination from air and boning hall surfaces in the plant. There are opportunities for the lamb industry to have a more consistent shelf life by reducing the variability within and between slaughter batches for all the microbiological and quality attributes measured. Measures to support an extended shelf life of VP lamb include trying to achieve lower microbial counts at the time of packaging, use of packaging film with low oxygen permeability and rigorous temperature control, combined with lowering of the temperature during VP storage.

## Figures and Tables

**Figure 1 foods-11-01850-f001:**
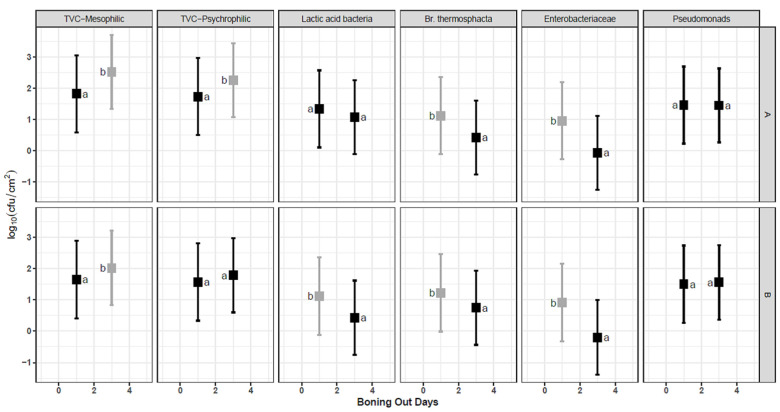
Microbial count’s (log_10_ cfu cm^−2^) estimated mean ± standard error on boning hall surfaces when lamb carcasses of Clean livestock Policy (CLP) A or B were boned-out following chilling for 24 h (D1, T1) or 72h (D3, T2). The error bars indicate 95% CI intervals for the replicates (*n* = 20). Different colours and lowercase letters a and b indicate significant differences (*p* < 0.05) with Tukey adjusted post hoc comparisons. Uppercase letters indicate CLP A and B.

**Figure 2 foods-11-01850-f002:**
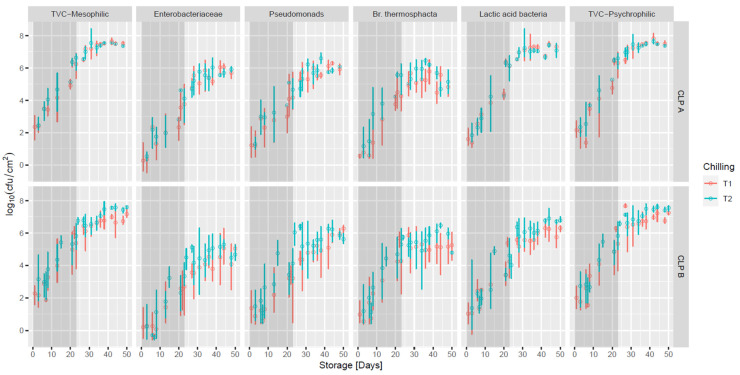
Microbial counts (log_10_ cfu cm^−2^) estimated mean and standard error (±) on VP lamb joints from carcasses (*n* = 237) chilled for 24 h (T1) or 72 h (T2) from lambs of Clean Livestock Policy (CLP) categories A and B. Darker blocks represent storage of lamb joints at the lamb plant (D1 to D23 at 1.7 ± 1.6 °C), and light grey blocks represent retail display storage (D23 to D50 at 4.6 ± 1.8 °C).

**Figure 3 foods-11-01850-f003:**
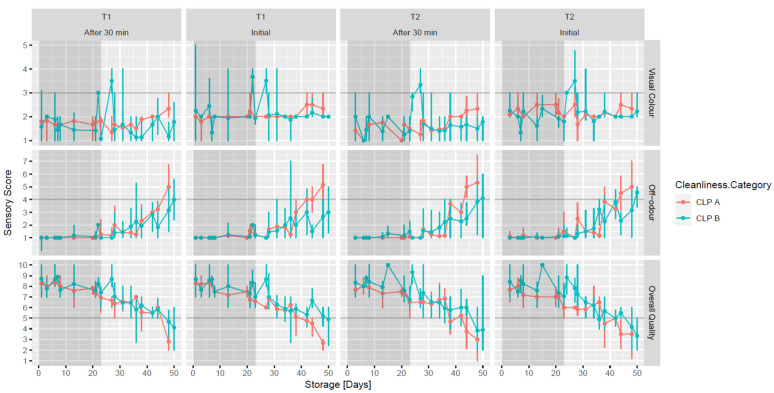
Raw meat quality assessment estimated mean and standard error (±) of visual colour, off-odour and overall quality of VP lamb joints when tested initially and 30 min after opening, which were derived from carcasses chilled for 24 h (T1) or 72 h (T2) and scored as Clean livestock Policy (CLP) A or CLP B. Visual colour which scored ≥3 (scale from 1 to 5), off-odour which scored ≥4 (scale from 1 to 8) and overall quality of the meat which scored ≤5 (scale from 10 to 1) were deemed to be unacceptable.

**Figure 4 foods-11-01850-f004:**
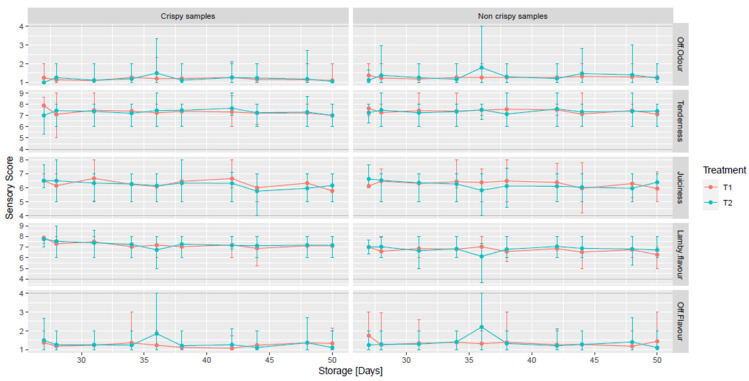
Eating quality assessment (off-odour, tenderness, juiciness, lamb flavour and off-flavour) of vacuum-packed lamb joints derived from carcasses chilled for 24 h (T1) or 72 h (T2) over 50 days chilled storage. Samples were prepared as crispy (the outer seared area) or non-crispy (inner area of a joint), respectively. Off-odour and off-flavour that scored ≥4 were deemed to be unacceptable (5-point category scale from 1 to 5). Tenderness, juiciness and lamb flavour was scored using a 10-point numerical scale from 1 to 10.

**Table 1 foods-11-01850-t001:** Fleece microbial counts’ (log_10_ cfu cm^−2^) estimated mean ± standard error at two abattoirs based on mean data from slaughter batches of Clean Livestock Policy (CLP) score A or B, which used clipping (abattoir 1) (*n* = 3) and no clipping (abattoir 2) (*n* = 2).

Microorganisms	CLP Category	Clipping (Abattoir 1)	No Clipping (Abattoir 2)	Clipping V No Clipping (*)
^1^ TVC-Mesophilic	A	4.96 ± 0.09 ^y^	5.78 ± 0.12 ^a^	*
	B	4.16 ± 0.07 ^x^	5.97 ± 0.13 ^a^	*
^1^ TVC-Psychrophilic	A	4.60 ± 0.09 ^y^	5.54 ± 0.12 ^a^	*
	B	3.77 ± 0.07 ^x^	5.77 ± 0.13 ^a^	*
Lactic acid bacteria	A	3.50 ± 0.09 ^y^	3.33 ± 0.13 ^a^	
	B	1.97 ± 0.07 ^x^	3.36 ± 0.12 ^a^	*
*Br. thermosphacta*	A	1.55 ± 0.09 ^y^	0.83 ± 0.12 ^a^	*
	B	1.13 ± 0.07 ^x^	2.66 ± 0.13 ^b^	*
*Enterobacteriaceae*	A	1.78 ± 0.09 ^y^	1.18 ± 0.12 ^a^	
	B	−0.01 ± 0.07 ^x^	1.97 ± 0.13 ^b^	*
Pseudomonads	A	2.29 ± 0.09 ^y^	2.49 ± 0.12 ^a^	*
	B	0.88 ± 0.07 ^x^	2.80 ± 0.13 ^a^	*

^1^ TVC = Total viable Count; Letters x and y display post hoc comparison tests of microbial counts (log_10_ cfu cm^−2^) between CLP Categories for each microorganism at abattoir 1 (clipping). Letters a and b display post hoc comparison tests of microbial counts (log_10_ cfu cm^−2^) between CLP Categories for each microorganism at abattoir 2 (no clipping). * Indicates a significant difference in microbial counts (log_10_ cfu cm^−2^) between clipping (abattoir 1) and no clipping (abattoir 2) for each CLP Category and each microorganism based on a post hoc comparison test.

**Table 2 foods-11-01850-t002:** Aerial microbial counts’ (log_10_ cfu m^−2^h^−1^) estimated mean ± standard error in the fleece pull area at two abattoirs based on mean data from slaughter batches of Clean Livestock Policy (CLP) score A or B, which used clipping (abattoir 1) (*n* = 3) and no clipping (abattoir 2) (*n* = 2).

Microorganisms	CLP Category	Clipping (Abattoir 1)	No Clipping (Abattoir 2)
^1^ TVC-Mesophilic	A	4.49 ± 0.61 ^a^	TNTC ^2^
	B	4.60 ± 0.43 ^a^	TNTC
^1^ TVC-Psychrophilic	A	4.27 ± 0.43 ^a^	TNTC
	B	4.43 ± 0.30 ^a^	TNTC
Lactic acid bacteria	A	3.37 ± 0.35 ^a^	TNTC
	B	3.80 ± 0.43 ^a^	TNTC
*Br. thermosphacta*	A	3.03 ± 0.30 ^a^	TNTC
	B	3.67 ± 0.43 ^a^	TNTC
*Enterobacteriaceae*	A	2.20 ± 0.43 ^a^	TNTC
	B	2.44 ± 0.43 ^a^	TNTC
Pseudomonads	A	2.85 ± 0.43 ^a^	TNTC
	B	2.93 ± 0.30 ^a^	TNTC

^1^ TVC = Total viable Count; ^2^ TNTC = colonies too numerous to count. ^a^ Different superscript lowercase letters in a column indicate significant differences (*p* < 0.05) between CLP categories for each individual microorganism.

**Table 3 foods-11-01850-t003:** Microbial counts; (log_10_ cfu cm^−2^) estimated mean ± standard error on derived pre-chill and post-chill carcasses from lamb of Clean Livestock Policy (CLP) score A or B, at T1 (D1, post-slaughter) or T2 (D3, post-slaughter) based on mean data from all slaughter batches (*n* = 5).

Microorganisms	CLP Category	Pre-Chill T1	Pre-Chill T2	Post-Chill T1	Post-Chill T2
^1^ TVC- Mesophilic	A	2.54 ± 0.30 ^x^	2.54 ± 0.29 ^x^	2.46± 0.37 ^ax^	2.43 ± 0.38 ^ax^
	B	2.75 ± 0.25 ^x^	2.68 ± 0.23 ^x^	2.37± 0.31 ^ax^	2.48 ± 0.31 ^ax^
^1^ TVC-Psychrophilic	A	2.07 ± 0.30 ^x^	2.03 ± 0.29 ^x^	1.92± 0.37 ^ax^	2.01 ± 0.38 ^ax^
	B	2.23 ± 0.25 ^x^	2.22 ± 0.23 ^x^	1.92± 0.31 ^ax^	2.25 ± 0.31 ^bx^
Lactic acid bacteria	A	0.95 ± 0.30 ^x^	0.93 ± 0.29 ^x^	1.05± 0.37 ^ax^	1.26 ± 0.38 ^ax^
	B	1.26 ± 0.25 ^x^	0.94 ± 0.23 ^x^	0.86± 0.31 ^bx^	0.37 ± 0.31 ^ax^
*Br. thermosphacta*	A	0.47 ± 0.30 ^x^	0.71 ± 0.29 ^x^	0.26± 0.37 ^ax^	0.13 ± 0.38 ^ax^
	B	−0.10 ± 0.25 ^x^	−0.02 ± 0.23 ^x^	−0.11 ± 0.31 ^ax^	0.51 ± 0.31 ^bx^
*Enterobacteriaceae*	A	1.24 ± 0.30 ^y^	1.23 ± 0.29 ^y^	0.53 ± 0.37 ^ax^	0.41 ± 0.38 ^ax^
	B	0.00 ± 0.25 ^x^	−0.12 ± 0.23 ^x^	−0.83 ± 0.31 ^bx^	−1.04 ± 0.31 ^ax^
Pseudomonads	A	1.40 ± 0.30 ^x^	1.49 ± 0.29 ^x^	1.14 ± 0.37 ^ax^	1.14 ± 0.38 ^ax^
	B	0.63 ± 0.25 ^x^	0.44 ± 0.23 ^x^	0.22 ± 0.31 ^ax^	0.72 ± 0.31 ^bx^

^1^ TVC = Total viable Count; Letters x and y display post hoc comparison tests of microbial counts (log_10_ cfu cm^−2^) between CLP categories at pre-chill and post-chill times for each microorganism. Letters a and b display post hoc comparison tests of microbial counts (log_10_ cfu cm^−2^) between T1 and T2 at the post-chill time for each microorganism and CLP Category.

**Table 4 foods-11-01850-t004:** Aerial microbial counts; (log_10_ cfu m^−2^h^−1^) estimated mean ± standard error in the evisceration area (D0), chill room (D0, D1 and D3 post-slaughter) and boning hall (D1 and D3 post-slaughter) when processing lambs of Clean Livestock Policy (CLP) score A or B.

Microorganisms	CLP	Evisceration Area	Chill Room D0	Chill Room D1	Chill Room D3	Boning Hall D1	Boning Hall D3
^1^ TVC Mesophilic	A	4.48 ± 0.28 ^x^	3.08 ± 0.23 ^b^	2.35 ± 0.23 ^a^	3.03± 0.23 ^ab^	3.25 ± 0.28 ^r^	3.48 ± 0.28 ^r^
	B	5.02 ± 0.25 ^x^	3.33 ± 0.19 ^a^	3.69 ± 0.19 ^a^	3.15 ± 0.23 ^a^	3.58 ± 0.24 ^r^	3.70 ± 0.25 ^r^
^1^ TVC Psychrophilic	A	3.98 ± 0.24 ^x^	2.87 ± 0.23 ^a^	2.57 ± 0.23 ^a^	2.90 ± 0.23 ^a^	2.64 ± 0.28 ^r^	3.02 ± 0.28 ^r^
	B	4.64 ± 0.20 ^y^	3.17 ± 0.19 ^a^	3.03 ± 0.19 ^a^	2.84 ± 0.23 ^a^	3.09 ± 0.23 ^r^	3.00 ± 0.25 ^r^
Lactic acid bacteria	A	3.58 ± 0.24 ^x^	2.60 ± 0.23 ^a^	2.44 ± 0.23 ^a^	2.50 ± 0.23 ^a^	2.59 ± 0.28 ^r^	2.69 ± 0.28 ^r^
	B	3.63 ± 0.20 ^x^	2.56 ± 0.19 ^a^	2.48 ± 0.19 ^a^	2.60 ± 0.23 ^a^	3.49 ± 0.23 ^s^	2.66 ± 0.25 ^r^
*Br. thermosphacta*	A	2.57 ± 0.24 ^x^	2.20 ± 0.23 ^a^	2.20 ± 0.23 ^a^	2.20 ± 0.23 ^a^	2.20 ± 0.28 ^r^	2.27 ± 0.28 ^r^
	B	2.50 ± 0.20 ^x^	2.20 ± 0.19 ^a^	2.20 ± 0.19 ^a^	2.18 ± 0.23 ^a^	2.53 ± 0.23 ^r^	2.43 ± 0.25 ^r^
*Enterobacteriaceae*	A	3.33 ± 0.24 ^x^	2.20 ± 0.23 ^a^	2.20 ± 0.23 ^a^	2.20 ± 0.23 ^a^	2.20 ± 0.28 ^r^	2.20 ± 0.28 ^r^
	B	2.85 ± 0.20 ^x^	2.20 ± 0.19 ^a^	2.20 ± 0.19 ^a^	2.18 ± 0.23 ^a^	2.87 ± 0.23 ^s^	2.43 ± 0.25 ^r^
Pseudomonads	A	3.09 ± 0.24 ^x^	2.20 ± 0.23 ^a^	2.50 ± 0.23 ^a^	2.54 ± 0.23 ^a^	2.20 ± 0.28 ^r^	2.57 ± 0.28 ^r^
	B	3.18 ± 0.20 ^x^	2.44 ± 0.19 ^a^	2.20 ± 0.19 ^a^	2.18 ± 0.23 ^a^	2.25 ± 0.23 ^r^	2.51 ± 0.25 ^r^

^1^ TVC = Total viable Count; Letters x and y display post hoc comparison tests of microbial counts (log_10_ cfu m^−2^h^−1^) between CLP Categories at the evisceration area for each microorganism. Letters a and b display post hoc comparison tests of microbial counts (log_10_ cfu m^−2^h^−1^) between days in the chill room for each microorganism and CLP Category. Letters r and s display post hoc comparison tests of microbial counts (log_10_ cfu m^−2^h^−1^) between days in the boning hall for each microorganism and CLP Category.

## Data Availability

Not applicable.

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
