# Peer review of "Impact of Industrial Practices on the Microbial and Quality Attributes of Fresh Vacuum-Packed Lamb Joints"

_foods, 2022, doi:10.3390/foods11131850_

Round 1
Reviewer 1 Report
I tried hard but could not find any flaw, this paper is absolutely flawless
Author Response
The authors acknowledge and thank the reviewer for this kind comment
Reviewer 2 Report
The manuscript entitled by De Alba et al. "Impact of industrial practices on the microbial and quality attributes of fresh vacuum packed lamb joints" is extensive and comprehensive scientific research, that deals with the impact of different industrial practices of abattoirs on the several aspects quality of lamb meat during storage periods. Such extensive work could be having negative aspects when be shown and presented in one manuscript with restricted pages such as not able to show all results (e.g. TBARS) or show the results intensively and crowded (e.g. Figure 2) or inability to discuss the results adequately and extensively. This, in turn, creates difficulties for the reader to understand and assimilate.
However, there are some issues need to be considered:
Line 153: you indicated "the back legs were boned-out into bone-in joints" explain pleases what you mean about bone-in joints. Also mention the anatomical names of the joins
Line 290-292: you indicated "However, during this trial period CLP C scored lambs were not being accepted for slaughter, and so this will have mitigated the impact of the dirtiest lambs entering the plants, though this could not be directly measured in the study" On what basis CLP C scored lambs was excluded from the study without directly measured?
Table 2: TNTC is abbreviation of what?
Line 315-316: you indicated "It is notable that the aerial microbial counts in the fleece pull area (Table 2) in abattoir 1, where fleece clipping was carried out, were also lower than in abattoir 2 where no clipping was performed" but there are no data for No Clipping (Abattoir 2) in table 2 to confirm that.
Table 3: there is addition ''y'' letter for result of Enterobacteriaceae, B CLP Category, Pre-Chill T2 ''−0.12±0.23x y''
Line 344: delete the full stop from the sentence
Line 346: add the full stop to the sentence
Author Response
The authors thank the reviewer for the comments and the replies are outlined below
1. Line 153: you indicated "the back legs were boned-out into bone-in joints" explain pleases what you mean about bone-in joints. Also mention the anatomical names of the joints
The authors apologise for the lack of clarity. The lamb leg is a primal cut which may be sold to the consumer, with the bone still in the leg or the bone may be removed during the cutting up the carcass. The presence/ absence of the bone will impact on the microbiology and shelf life of the meat during storage. The hind “lamb leg” is a generally recognisable name for this primal cut which includes the bone and groups of muscles including biceps femoris, semitendinosus, semimembranosus, vastus lateralis, tensor fascia lata, and gluteus medius. The text is modified to clarify that it is lamb hind leg that was studied, but do not consider it is useful to name the muscles as they were not dissected and individually studied in the trials.
2. Line 290-292: you indicated "However, during this trial period CLP C scored lambs were not being accepted for slaughter, and so this will have mitigated the impact of the dirtiest lambs entering the plants, though this could not be directly measured in the study" On what basis CLP C scored lambs was excluded from the study without directly measured?
The CLP scoring is a visual assessment (score) of the level of dirt on the lamb fleece. The assessment is carried out on arrival of the lamb into the lairage area. During the study period any lamb that scored a “C” were not allowed to proceed to slaughter, they had to be returned to the farm and so animals scoring a “C” were not included in our trial. The text is amended in section 2.1 and 3.1 to clarify this.
3. Table 2: TNTC is abbreviation of what?
TNTC, means “too numerous to count”, A foot note is added to the table
4. Line 315-316: you indicated "It is notable that the aerial microbial counts in the fleece pull area (Table 2) in abattoir 1, where fleece clipping was carried out, were also lower than in abattoir 2 where no clipping was performed" but there are no data for No Clipping (Abattoir 2) in table 2 to confirm that.
Apologies for lack of clarity. In abattoir 2, the TNTC result, now clarified, indicates that the number of colonies were too numerous to count, and thus shows that the aerial microbial counts were all significantly higher in abattoir 2 than in abattoir 1 where a count was achievable (lower number of colonies and thus readable).
The manuscript text is clarified to explain this better
5. Table 3: there is addition ''y'' letter for result of Enterobacteriaceae, B CLP Category, Pre-Chill T2 ''−0.12±0.23x y''
This error is corrected
6. Line 344: delete the full stop from the sentence
This is done
7. Line 346: add the full stop to the sentence
This is done
Reviewer 3 Report
Well written paper of scientific significance
Author Response
The authors thank the reviewer for their kind comment